# Response of Tallgrass Prairie to Management in the U.S. Southern Great Plains: Site Descriptions, Management Practices, and Eddy Covariance Instrumentation for a Long-Term Experiment

**Pradeep Wagle \***, **Prasanna H. Gowda**, **Brian K. Northup, Patrick J. Starks** and **James P. S. Neel**

USDA, Agricultural Research Service, Grazinglands Research Laboratory, El Reno, OK 73036, USA
* Correspondence: pradeep.wagle@usda.gov

**Abstract:** Understanding the consequences of different management practices on vegetation phenology, forage production and quality, plant and microbial species composition, greenhouse gas emissions, and water budgets in tallgrass prairie systems is vital to identify best management practices. As part of the Southern Plains Long-Term Agroecosystem Research (SP-LTAR) grassland study, a long-term integrated Grassland-LivestOck Burning Experiment (iGLOBE) has been established with a cluster of six eddy covariance (EC) systems on differently managed (i.e., different burning and grazing regimes) native tallgrass prairie systems located in different landscape positions. The purpose of this paper is to describe this long-term experiment, report preliminary results on the responses of differently managed tallgrass prairies under variable climates using satellite remote sensing and EC data, and present future research directions. In general, vegetation greened-up and peaked early, and produced greater forage yields in burned years. However, drought impacts were greater in burned sites due to reductions in soil water availability by burning. The impact of grazing on vegetation phenology was confounded by several factors (e.g., cattle size, stocking rate, precipitation). Moreover, prairie systems located in different landscapes responded differently, especially in dry years due to differences in water availability. The strong correspondence between vegetation phenology and eddy fluxes was evidenced by strong linear relationships of a greenness index (i.e., enhanced vegetation index) with evapotranspiration and gross primary production. Results indicate that impacts of climate and management practices on vegetation phenology may profoundly impact carbon and water budgets of tallgrass prairie. Interacting effects of multiple management practices and inter-annual climatic variability on the responses of tallgrass prairie highlight the necessity of establishing an innovative and comprehensive long-term experiment to address inconsistent responses of tallgrass prairie to different intensities, frequencies, timing, and duration of management practices, and to identify best management practices.

**Keywords:** global positioning system; interacting effects; remote sensing; unmanned aerial systems; vegetation phenology

## 1. Introduction

Although the coverage of tallgrass prairies once occupied $>68 \times 10^6$ ha of the North American Great Plains before European Settlement, the remaining 1% [1], after mostly converted to croplands, is still important for livestock production in several states of the United States. Some tallgrass prairies are unmanaged as low input production systems, while others are highly managed, which is mainly driven by economics and need of farmers/ranchers. In addition, these grasslands frequently experience

different disturbances and management practices such as burning, grazing, drought, fertilization, and harvesting of hay [2]. Prescribed burning is a widely recognized, necessary, and an important management practice for prairie systems because properly applied prescribed burns can stimulate plant production, improve forage quality, and control weeds and woody encroachments [3]. Understanding the consequences of different management practices and disturbances is vital to identify and adopt best management practices [2]. However, the response of tallgrass prairie to management practices and disturbances might be influenced by other treatments and/or management practices (e.g., fertilization, burning, grazing), their timing and duration, and other factors such as differences in plant communities (i.e., the proportion of $C_3$ and $C_4$ species) and landscape positions [4]. As such, previous studies have reported inconsistent responses of tallgrass prairie to different intensities, frequencies, timing, and duration of management practices and disturbances [2,5–7]. Thus, the responses of tallgrass prairie to main and interacting effects of a suite of management practices should be examined over longer (a decade or more) periods as most previous studies were short-term (2–4 years).

Climatic gradients can have confounding effects on the responses of agroecosystems to management practices. Therefore, evaluating the responses of the same biome or land use type to different management practices within the same climatic condition is required to investigate the effects of non-climatic factors and management practices [8]. Eddy covariance (EC) techniques have been widely used to study the exchange of energy, carbon dioxide ($CO_2$), and water vapor ($H_2O$) fluxes between agroecosystems and the atmosphere over the past two decades [9]. It is also well documented that management practices and disturbances influence the dynamics of $CO_2$, $H_2O$, and surface energy fluxes [10–13]. Although several studies have reported $CO_2$ and $H_2O$ dynamics in tallgrass prairie, most have utilized one or a small number of EC systems to examine individual forms of management [13–18]. If the spatiotemporal variability of $CO_2$ and $H_2O$ fluxes and the underlying mechanisms can be better characterized, then such information can be extrapolated and applied over broader areas and longer periods [19,20]. However, comparative studies of $CO_2$ and $H_2O$ fluxes in tallgrass prairie from different landscape positions (e.g., upland, intermediate, and lowland) that experience different management practices (e.g., grazing, spring burns, haying) under the same climatic regime is scarce. Thus, there is a significant need to use multiple EC systems in co-located tallgrass prairie pastures to examine their responses to different frequencies and timing of spring burns and grazing regimes under similar environmental conditions.

To address this knowledge gap, a long-term "integrated Grassland-LivestOck Burning Experiment (iGLOBE)" has been established, with a cluster of six EC systems (Figure 1) deployed on a series of tallgrass prairie pastures with different management practices (i.e., different burning and grazing regimes), at the United States Department of Agriculture-Agricultural Research Service (USDA-ARS), Grazinglands Research Laboratory (GRL), in central Oklahoma, USA. These prairie systems represent different positions within the landscape of the watershed of a local stream, ranging from upper most ridge and riser positions to lowest tread–lowland. This innovative and comprehensive long-term experiment was established with the following objectives:

i. Establish a cluster of EC towers within a suite of tallgrass prairie pastures to develop long-term databases of surface energy, $CO_2$, and $H_2O$ budgets along with plant biometric measurements and climate data.

ii. Compare carbon and water dynamics/budgets, and vegetation phenology in tallgrass prairie under combinations of prescribed spring burns and grazing regimes in different landscape positions under a variable climate.

iii. Understand variability in forage production and quality, macronutrient availability, soil and landscape features, cattle grazing behavior, and forage utilization within prairie systems using geospatial techniques and sensors.

The purpose of this paper is to describe this long-term experiment and report preliminary results on the responses of differently managed tallgrass prairie pastures under a variable climate. The long-term

goal of the experiment is to improve our understanding of the impact of management practices on vegetation phenology, and carbon and water dynamics in tallgrass prairie, which will foster flexible and resilient forage-livestock systems to meet growing demands of feed, fiber, and food.

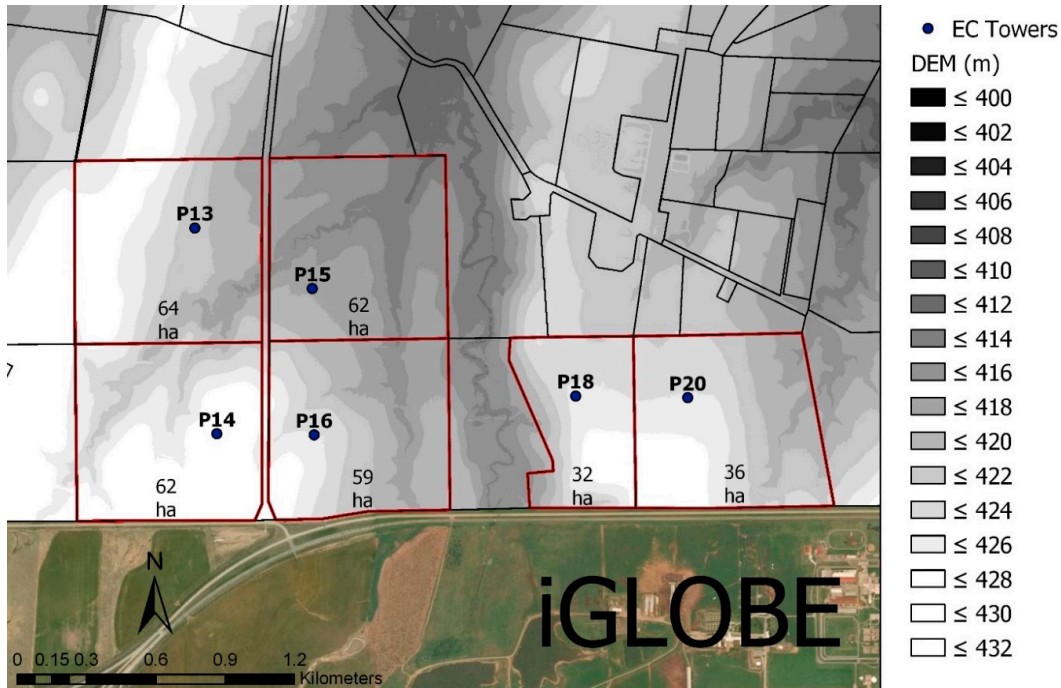

**Figure 1.** Layout of experimental tallgrass prairie pastures at the United States Department of Agriculture-Agricultural Research Service (USDA-ARS), Grazinglands Research Laboratory (GRL), El Reno, Oklahoma. The red borders represent the experimental pastures used in this study.

## 2. Materials and Methods

### 2.1. Experimental Layout and Site Description

The experimental area is located on a rolling upland landscape that contains parts of watersheds of local streams and includes a range of landscape features. These features are easterly and westerly-facing slopes of 3 to 6% on riser positions, and toe and tread slope positions with 0 to 2% slope, bordering the risers. The experiment includes three different replicate areas of southern tallgrass prairie. One area is composed of four pastures (P13, P14, P15, and P16) that are components of a 247 ha area managed to support a beef cow herd. A second 32 ha area (P18) is divided into nine paddocks and grazed by yearling stocker cattle during the early growing season (May–July). A third 36 ha area (P20) is managed to provide high quality hay, harvested in early July.

Plant communities of these sites are mixes of native species, indicative of rangeland sites in good condition [21]. The dominant species are the warm-season grasses—big bluestem [*Andropogon gerardii* Vitman], Indiangrass [*Sorghastrum nutans* (L.) Nash], and little bluestem [*Schizachryium scoparium* (Michx.) Nash]. All pastures except P20 are located within the same watershed (Figure 1), occupying a series of landscape positions—ranging from upper-most ridge and upper riser (P14) to the lower tread-lowland areas adjacent to the stream (P15). The landscape features and the area covered by one Moderate Resolution Imaging Spectroradiometer (MODIS) pixel (~500 m resolution) in each pasture are shown in Figure 2.

A range of soils that belong to different families and subgroups, based on landscape position, of the Mollisol order were recorded in the area. All listed soils evolved from parent material derived from the Dog Creek formation, a reddish-brown shale containing thin inter-beds of sandstone and siltstone [22]. Three sub-types of Norge series silt loams (Fine-silty, mixed, active thermic Udic Paleustolls), situated

on riser (mid-slope) positions of the landscape, are the most-common. Kirkland or Renfrow silt loams (Fine, mixed superactive, thermic Udertic Paleustolls) situated on tread positions and Port silt loams (Fine-silty, mixed superactive, thermic Cumulic Haplustolls) at toe positions bound the Norge series.

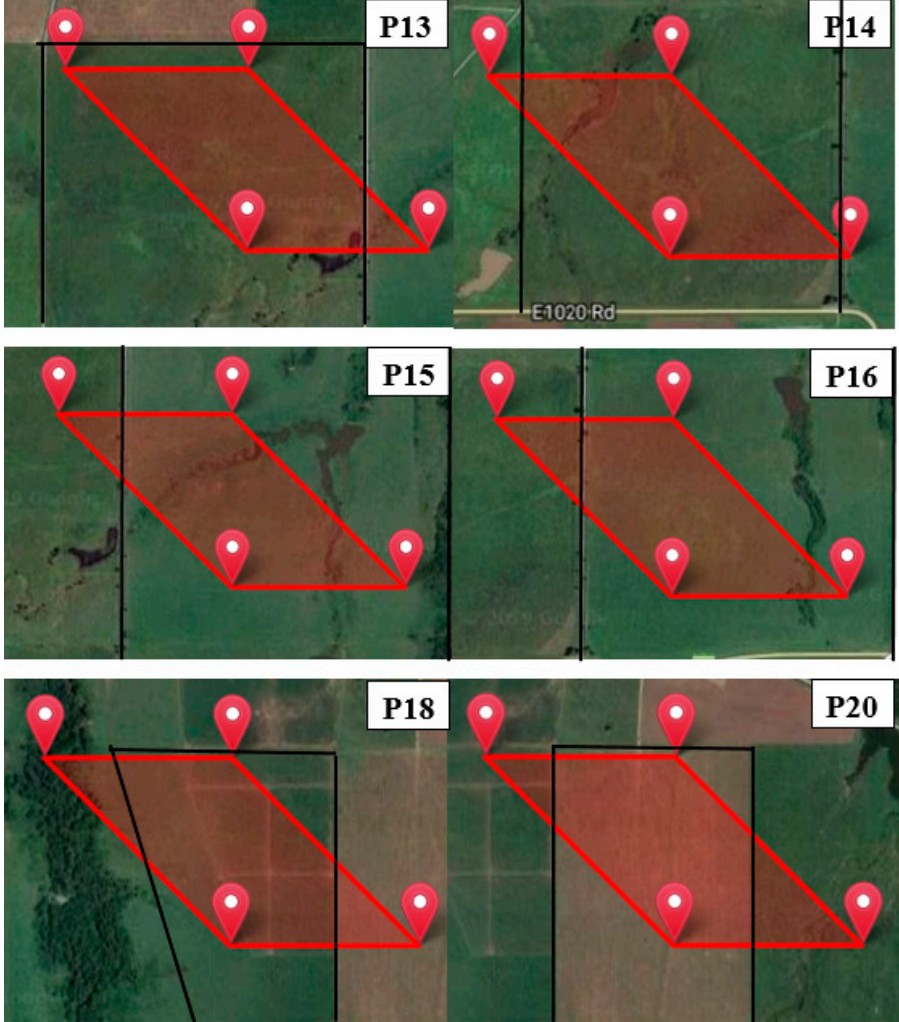

**Figure 2.** Landscape features of pastures. The red borders represent the size of one Moderate Resolution Imaging Spectroradiometer (MODIS) pixel at ~500-m spatial resolution and black lines represent the boundary of the pastures.

Six additional related soils exist as inclusions or complexes within boundaries of each of the primary soil series. The upper 30 cm of the profile of these series have near-neutral pH of 6.7, cation exchange capacity of 13.5 cmol·kg$^{-1}$ soil, low water-holding capacity of 3 mm·cm$^{-1}$ soil, and permeability rate of 33 mm·h$^{-1}$. Details of each pasture are presented below.

### 2.1.1. Pastures 13–16

Pastures 13, 14, 15, and 16 are managed as a single group that are in a rotation of yearlong grazing by 50 cow-calf pairs. The cattle are grazed in these pastures for 30-day periods, separated by 90-day rest periods. These pastures receive a prescribed spring burn in mid to late-February on a 4-year rotation, as part of their normal management. Years of burning for the last decade are summarized in Table 1. The timing of these grazing-rest rotations changes annually, with all periods reset to May 1 for the pastures receiving a prescribed burn that year. This approach allows an adaptive form of management to all pastures, based on changes in number, timing, frequency, and intensity of grazing bouts, in a repeatable (and replicable) fashion over the 4-year cycle.

**Table 1.** Years of burning in the last decade, landscape positions, and the periods of eddy covariance (EC) measurement for the tallgrass prairie pastures.

| Pastures | Burned Years | Landscape Position | EC Measurement |
|---|---|---|---|
| P13 | 2013, 2018 | Intermediate–toe through tread | 2013–Current |
| P14 | 2014, 2019 | Upper most–tread and upper riser | May 2019–Current |
| P15 | 2012, 2017 | Lowest toe–lowland along stream | July 2019–Current |
| P16 | 2011, 2015 | Intermediate–upper riser to upper toe | May 2019–Current |
| P18 | Each year | Toe through tread | May–August 2016 June 2017–Current |
| P20 | Each year | Riser through tread | April 2019–Current |

The cow herd splits during breeding seasons (June–July) with animals assigned to two different pastures of the rotation, based on weight and frame size (medium, and large) of cows. Roughly 1500 animal-unit-days (AUD) of grazing pressure are applied per 'normal' grazing bout, or ~4500 AUD per year, assuming three bouts per year. This equates to ~61,000 kg forage used per pasture per year, assuming an allotment of 13.6 kg forage per head per day. This translates to ~18,250 AUD grazing per year for the entire set of paddocks, or ~1 Mg·ha$^{-1}$ forage removal per year. This level of grazing pressure was chosen as dry years may only supply 1 Mg·ha$^{-1}$ forage for grazing, assuming total production of ~2 Mg·ha$^{-1}$ so that enough plant biomass was left to ensure health of the landscape. An extra pasture is included in management of these pastures to provide an extra area to aid in splitting into two herds during the breeding season, and for holding animals in February to defer the pasture being burned. Broadleaf herbicides are applied to burned pastures during late-April of the burned year and the following year to control a series of weeds [annual sunflower (*Helianthus annuus* L.), mare's tail (*Conyza canadensis* (L.) Crong), and musk thistle (*Carduus nutans* L.)]. The grazing schedule for pastures 13 through 16 is presented in Table 2.

**Table 2.** Planned timing of prescribed burn treatments and 30-day grazing (G) bouts for next four years for pastures 13 through 16.

| Year | Pasture | Jan | Feb | Mar | Apr | May | Jun | Jul | Aug | Sep | Oct | Nov | Dec |
|---|---|---|---|---|---|---|---|---|---|---|---|---|---|
| 2019 | P14 | | Burn | | | G | | | | G | | | |
| | P16 | | | | | | $\frac{1}{2}$ G | | | | G | | |
| | P15 | | | | | | | $\frac{1}{2}$ G | | | | G | |
| | P13 | | | | | | | G | | | | | G |
| 2020 | P14 | G | | | | | | | G | | | | G |
| | P16 | | Burn | | | G | | | | G | | | |
| | P15 | | | G | | | $\frac{1}{2}$ G | | | | G | | |
| | P13 | | | | G | | | $\frac{1}{2}$ G | | | | G | |
| 2021 | P14 | | | | G | | | $\frac{1}{2}$ G | | | | G | |
| | P16 | G | | | | | | | G | | | | G |
| | P15 | | Burn | | | G | | | | G | | | |
| | P13 | | | G | | | $\frac{1}{2}$ G | | | | G | | |
| 2022 | P14 | | | G | | | $\frac{1}{2}$ G | | | | G | | |
| | P16 | | | | | G | | $\frac{1}{2}$ G | | | | G | |
| | P15 | G | | | | | | | G | | | | G |
| | P13 | | Burn | | | G | | | | G | | | |

Note: Grazing periods are shifted to another extra pasture when assigned pasture to be burned. The timing of grazing reset to May 1 for the pasture receiving a prescribed burn that year.

### 2.1.2. Pasture 18

A 32 ha block of tallgrass prairie was divided into nine paddocks in 2012. Eight of these paddocks were assigned different intensive early stocking (IES) treatments, based on different lengths of grazing periods between mid-May through mid-July, ranging from 30 to 60-days (e.g., ~15 May to 15 June, 1 June to 30 June, 15 June to 15 July, 15 May to 15 July). These paddocks receive grazing deferments for the remainder of the year. Planned level of forage removal under IES treatments is ~1.5 Mg·ha$^{-1}$ on an annual basis. A ninth paddock is left ungrazed to serve as a control for comparisons. Pasture 18 is burned annually in spring.

### 2.1.3. Pasture 20

A 36 ha pasture of tallgrass prairie had received prescribed burns on a 3- to 5-year interval in mid-February during 1999 to 2015, and was harvested for hay in August. Since 2018, it has received annual prescribed spring burns in mid-February and has been harvested for hay in early-July, as an alternative form of intensive use during the early-growing season. Harvesting of hay removes a uniform segment of annual production compared to the patchier removal through grazing by cattle. These harvests leave 10–15 cm stubble height to match best management practices for haying, and to provide a comparison with grazing effects.

In addition to the above-mentioned pastures, we have been monitoring eddy fluxes of an alternative pasture (P11) of introduced grass since 2014, which offers additional comparisons with the differently managed native pastures. The pasture of introduced warm season grass was planted with old world bluestem (*Bothriochloa caucasica* C. E. Hubb.) in the mid 1980s. This pasture has received urea (40 to 80 kg N ha$^{-1}$) to meet the needs of different research projects over the years since establishment to support grazing by herds of cow-calf pairs. This pasture has also received herbicide treatments to control broadleaf weeds, and received the application of prescribed spring burns on a 3- to 5-year interval.

### 2.2. Burn Treatments

Prescribed early-spring burns are applied prior to initiation of growth of warm-season grasses (primarily mid- to late-February), with date of burning defined by weather conditions (wind speed and direction, air temperature, relative humidity, soil moisture) for safe burning. The objective of prescribed burns is to create head fires (moving downwind) that generate enough heat to consume all residual biomass. Such fires provide an additional benefit of improved control of weeds when combined with broadleaf herbicide in late-April of the burned years. Burning removes the effects of prior patch grazing in pastures, and supplies a uniform distribution of high quality forage for grazing by both stocker calves (P18) and cow-calf pairs (P13-P16), and high quality hay for feeding in winter (P20).

### 2.3. Eddy Covariance Measurements

A cluster of six EC systems has been established in tallgrass prairie pastures as a part of the GRL-FLUXNET, a network of 17 integrated flux measurement systems established at the USDA-ARS, GRL [8,23]. The EC systems are composed of an open path infrared gas analyzer (LI-7500 RS, LI–COR Inc., Lincoln, NE, USA) and a 3-D sonic anemometer (CSAT3, Campbell Scientific Inc., Logan, UT, USA). The EC systems collect $CO_2$, $H_2O$, and surface energy fluxes within the soil-plant-atmosphere complex. Additional meteorological variables include air temperature ($T_a$), relative humidity (RH), vapor pressure deficit (VPD), soil water content (SWC), soil temperature, soil heat flux, net radiation ($R_n$), and photosynthetic photon flux density (PPFD). In addition, P13 contains an Oklahoma Mesonet station as a part of more than 110 surface observing stations across Oklahoma for statewide monitoring of the mesoscale environment [24]. Precipitation data used in this study were obtained from the Oklahoma Mesonet website (http://mesonet.org/).

We computed 30-min values of eddy fluxes using the EddyPro software version 6.2.0 (LI-COR Inc.). Flux data flagged as poor quality (i.e., quality flags 2) and statistical outliers (>3.5 times of

standard deviation based on 2-weeks running window) were discarded [23]. Sensible (H) and latent (LE) heat fluxes beyond the ranges of −200 to 500 W m$^{-2}$ for H and −200 to 800 W m$^{-2}$ for LE were excluded [25,26]. The REddyProc package from the Max Planck Institute for Biogeochemistry, Germany (https://www.bgc-jena.mpg.de/bgi/index.php/Services/REddyProcWebRPackage) was used to fill gaps in data and to partition (nighttime-based) net ecosystem $CO_2$ exchange (NEE) into gross primary production (GPP) and ecosystem respiration (ER) [27,28].

### 2.4. Biometric Measurements

Leaf area index (LAI), percentage of canopy cover, canopy height, and biomass are being collected at monthly intervals during the active growing season. The LAI is measured using LAI-2200C plant canopy analyzer (LI-COR Inc., Lincoln, NE, USA). Percentage of canopy cover is measured using the Canopeo cell phone application (http://www.canopeoapp.com/). Biomass samples are collected destructively from $0.5 \times 0.5$ m$^2$ quadrats after measuring LAI and canopy cover. At each sampling, all biometric measurements are taken in five random locations along a 100 m transect in the north-south directions from the EC system. Fresh and dry (oven dried at 70 °C for a minimum of 48 h) biomass weights are measured.

## 3. Results and Discussion

### 3.1. Climatic Conditions

The climate is temperate continental, with a mean annual precipitation of ~900 mm and average temperature of ~15 °C [8]. Annual total precipitation ranged from <500 mm to >1300 mm in last two decades (Table 3), showing large year-to-year variability. This large degree of climatic (i.e., dry, normal, and wet years) variability offered a great opportunity to examine the responses of pastures to burning and grazing treatments. In the last decade, the years 2013, 2015, and 2017 received above long-term average precipitation, while other years received below long-term average precipitation.

**Table 3.** A summary of annual total precipitation, annual average air temperature ($T_a$), and total precipitation and average $T_a$ for the January–May period from 2000 to 2018.

| Year | Annual Total Precipitation (mm) | Annual Average $T_a$ (°C) | Jan–May Total Precipitation (mm) | Jan–May Average $T_a$ (°C) |
|---|---|---|---|---|
| 2000 | 893 | 15 | 358 | 11 |
| 2001 | 607 | 15 | 345 | 10 |
| 2002 | 791 | 14 | 266 | 10 |
| 2003 | 475 | 15 | 174 | 10 |
| 2004 | 849 | 15 | 202 | 11 |
| 2005 | 665 | 15 | 226 | 11 |
| 2006 | 629 | 16 | 229 | 13 |
| 2007 | 1359 | 15 | 540 | 10 |
| 2008 | 942 | 14 | 458 | 10 |
| 2009 | 795 | 14 | 297 | 10 |
| 2010 | 756 | 15 | 240 | 9 |
| 2011 | 642 | 16 | 162 | 10 |
| 2012 | 567 | 16 | 306 | 13 |
| 2013 | 1157 | 14 | 544 | 9 |
| 2014 | 610 | 14 | 154 | 9 |
| 2015 | 1273 | 15 | 661 | 10 |
| 2016 | 631 | 16 | 284 | 11 |
| 2017 | 1109 | 16 | 464 | 12 |
| 2018 | 795 | 15 | 147 | 10 |

### 3.2. Impact of Precipitation Distribution on Vegetation Phenology and Forage Production

Annual total precipitation may not be helpful to characterize differences in vegetation phenology and forage production, as distribution of precipitation during growing seasons is a key factor that affects vegetation phenology and productivity [29]. To illustrate differences in vegetation phenology and forage production with different distribution patterns of precipitation, we compared the dynamics of EVI in three years with lower annual total precipitation (2011, 2012, and 2014) for the pasture situated at the uppermost landscape position (P14). All three years received roughly two-third of the annual total precipitation. The spring was cooler and drier in 2011 and 2014, but warmer and wetter in 2012. Total precipitation for January–May was 162, 306, and 154 mm in 2011, 2012, and 2014, respectively. Average $T_a$ for January–May was 10, 13, and 9 °C in 2011, 2012, and 2014, respectively. As a result, prairie vegetation greened-up early (around mid-March) in 2012, but later (after mid-April) in 2011 and 2014 (Figure 3). The EVI peaked before mid-May 2012, while it peaked in early June in 2011 and early July in 2014. Peak EVI approximated 0.46, 0.52, and 0.72 in 2011, 2012, and 2014, respectively. Total precipitation received in June–July of 2011, 2012, and 2014 was 68, 58, and 260 mm, respectively. Due to severe dry and warm conditions, the EVI declined sharply after mid-June in 2011 and 2012. Due to higher precipitation in summer 2014, the EVI remained substantially higher throughout the growing season. The results illustrate the importance of timely distribution of precipitation during the growing season to productivity.

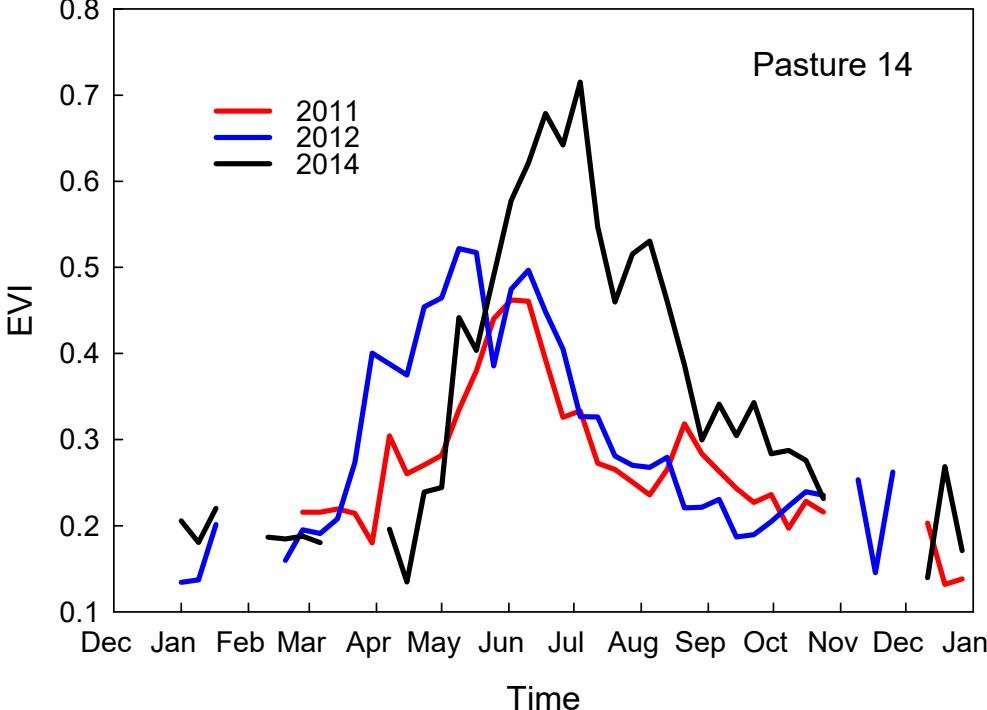

**Figure 3.** Seasonal dynamics of the enhanced vegetation index (EVI) in three selected years with different distribution patterns of precipitation at the uppermost landscape position (Pasture 14). The pasture was burned in 2014.

### 3.3. Impact of Burning on Vegetation Phenology

Pasture 14 was burned in 2014. As a result, the EVI showed earlier green-up and higher peak values (peak EVI >0.7) than the other pastures (peak EVI <0.6, Figure 4). Pastures 16, 18, and 20 were burned in 2015. As a result, the EVI showed earlier green-up and higher peak values at these pastures than the remaining pastures. In 2018, burning effects on early green-up and higher forage production did not appear due to dry conditions during spring. Total precipitation during January to May 2018

was only 147 mm (Table 3), the lowest for that period over the previous two decades. Consequently, vegetation phenology in all pastures was similar in 2018, regardless of burning treatment.

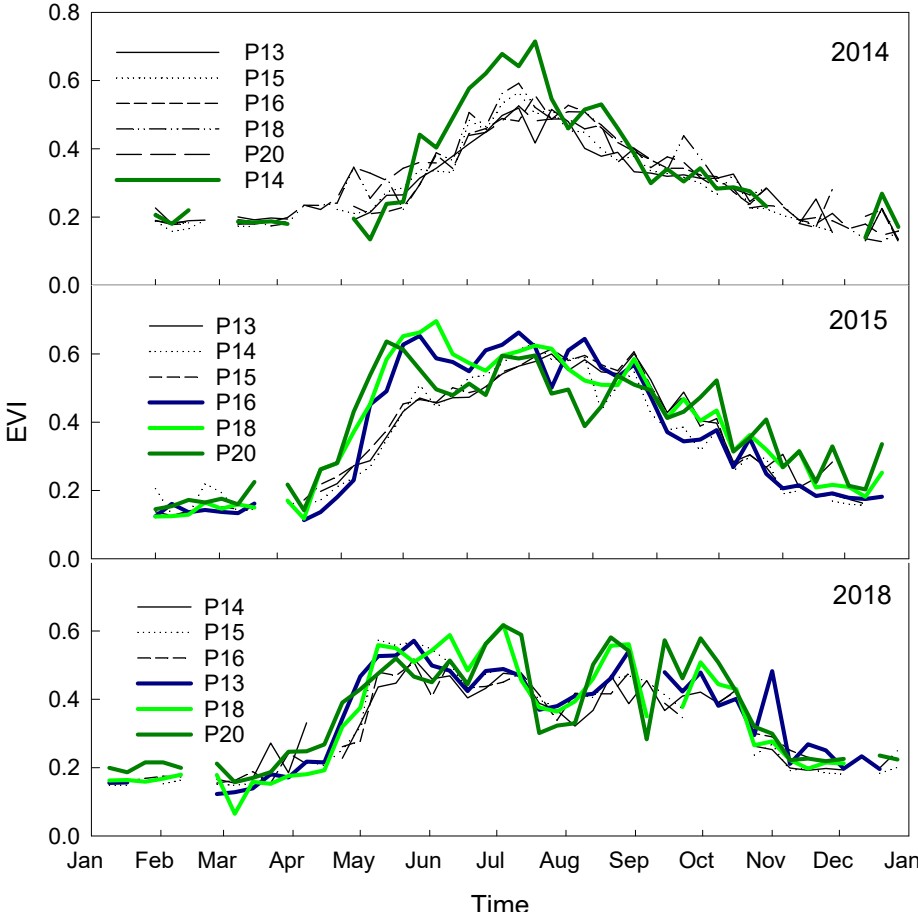

**Figure 4.** Seasonal dynamics of the enhanced vegetation index (EVI) at six tallgrass prairie pastures for the three selected years.

We compared the dynamics of EVI for P15 in dry (2012) and wet (2017) burned years vs. dry (2011) and wet (2015) unburned years to further examine the impacts of burning on vegetation phenology (Figure 5). Similarly, the dynamics of EVI were compared for P16 in dry (2011) and wet (2015) burned years vs. dry (2012) and wet (2017) unburned years. In general, vegetation greened-up and peaked earlier in burned years. Although 2012 was a dry year, the spring was warm and wet causing early green-up in 2012 in both pastures, though P16 was not burned in 2012. However, vegetation greened-up slightly earlier in the burned P15 than unburned P16. In comparison, both 2015 and 2017 were wet years and burning showed a clear impact on early green-up during both years. Vegetation greened-up and peaked early in 2017 (burned year) in P15 and 2015 (burned year) in P16. In P16, vegetation greened-up late and the EVI values were lowest during the drought-affected year (2011) despite application of a prescribed burn. The EVI values were greater in the unburned P15 than the burned P16 during the 2011 drought, illustrating negative impacts of the combination of a prescribed burn and drought.

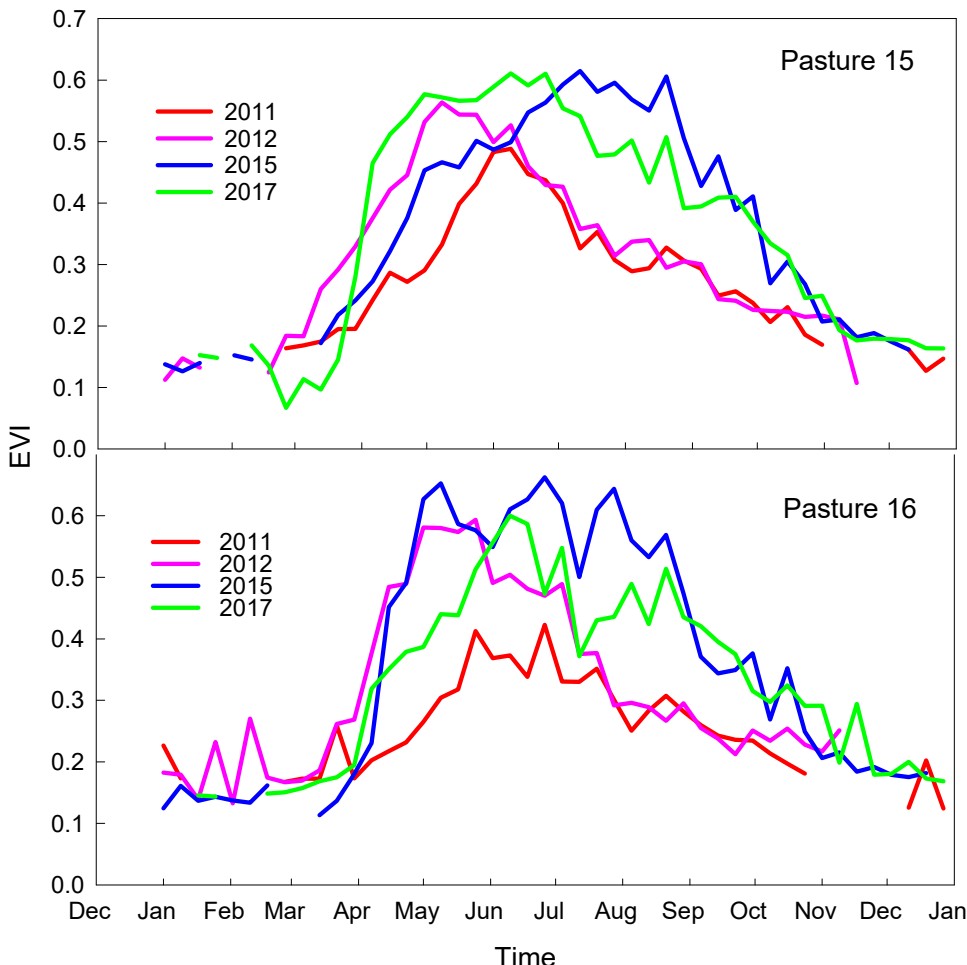

**Figure 5.** Seasonal dynamics of the enhanced vegetation index (EVI) in dry (2012) and wet (2017) burned years vs. dry (2011) and wet (2015) unburned years for pasture 15, and in dry (2011) and wet (2015) burned years vs. dry (2012) and wet (2017) unburned years in pasture 16.

In general, burning induces earlier green-up of vegetation due to greater soil heating and increased solar radiation at the soil surface by removing the litter layer [3]. We also observed early green-up of vegetation in burned years under normal climatic (i.e., no drought) conditions (Figure 4).

However, the burning had minimal or no impact on the timing of green-up in burned and unburned pastures in 2012 (Figure 6), when spring was warm (average $T_a$ was ~18 °C for March–May period as compared to long-term, 1981–2010, mean of ~15 °C) with well-distributed precipitation. These results indicate that greater temperature and availability of solar radiation at the soil surface are the main reasons for early green-up of vegetation following prescribed spring burns.

Burning also enhances below- and above-ground plant productivity in tallgrass prairie because of more plant-available nitrogen (N) due to increased mineralization from soil organic matter (SOM) as a consequence of higher soil temperature and additional impacts on biophysical properties and plant physiological responses [4,30,31]. We also observed relatively higher peak values of EVI during burned years under normal precipitation but not during dry years (Figure 4) due to reduced soil water availability. Burning can reduce availability of soil water in numerous ways: by increasing evaporation from bare soil immediately after burns, and by increasing plant transpiration due to earlier green-up and enhanced vegetative growth. Burning substantially reduced SWC at all depths up to 1.5 m in bluestem prairie in the Flint Hills, Kansas [32]. In comparison, soil water availability is higher in unburned sites because larger amounts of litter/mulch intercept more precipitation, increase infiltration of water into soil, and reduce soil evaporation [33]. Thus, the impact of drought on forage production

is greater on burned sites than unburned sites [34], as shown by smaller EVI in burned than unburned pasture during the droughts of 2011 and 2012 (Figure 6).

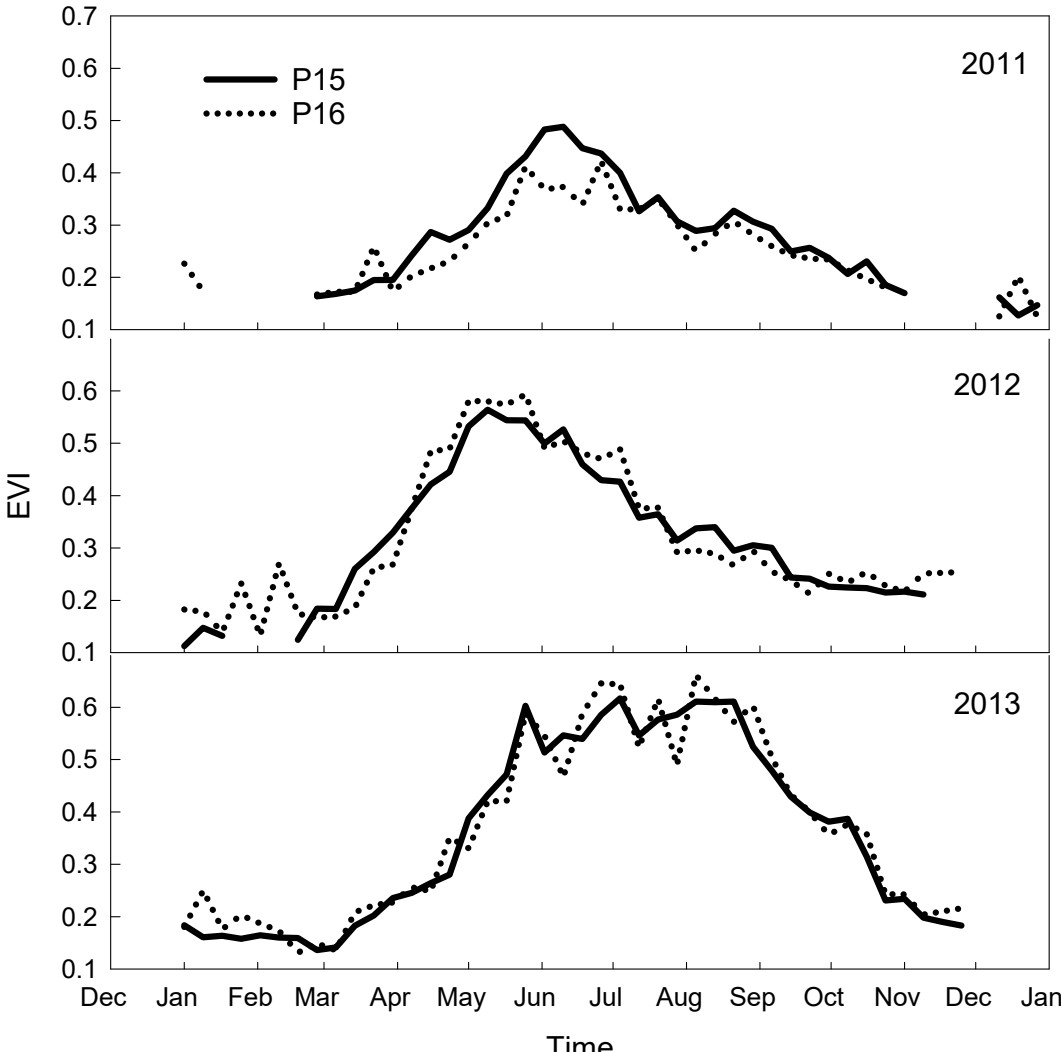

**Figure 6.** Seasonal dynamics of the enhanced vegetation index (EVI) for three selected years (dry years 2011 and 2012, and wet year 2013) in adjacent pastures located at different landscape positions. Pasture 15 (lowland along stream) was burned in 2012 and Pasture 16 (upper rise to toe) was burned in 2011.

*3.4. Impact of Landscape Positions on Vegetation Phenology*

Vegetation phenology was compared in adjacent pastures with different landscape positions (Figure 6). The EVI values were similar in a wet year 2013 in both pastures. Pasture 15 (lowest toe and lowland) was burned in 2012 and P16 (upper riser and toe) was burned in 2011. The unburned P16 greened-up as early as the burned P15 in 2012 due to warm temperatures combined with well-distributed precipitation during spring. Alternatively, burning did not induce early green-up in P16 during 2011, despite application of a prescribed burn. In 2011, both pastures greened-up around the same time and P15 (unburned pasture) had higher EVI values throughout the entire growing season, most likely because of greater availability of soil water due to differences in landscape position and burning treatment. Pasture 15 is mostly lowland along a stream and P16 has an intermediate landscape position with upper riser to upper toe. Thus, P15 was likely the recipient of overland flow of any runoff that occurred, and lateral flow through the soil profile from areas with higher elevation.

### 3.5. Comparisons of Haying vs. Grazing on Vegetation Phenology

Vegetation phenology was compared for two pastures (P15 and P20) with different management (i.e., grazing vs. haying, Figure 7). Haying is a form of intensive management that can be applied throughout growing seasons. In the current experiment, haying occurred during the early-growing season. As a result, there was an abrupt decline in EVI due to haying (P20) after hay harvest in early-July as compared to gradual decline in EVI due to rotational grazing in P15. Regrowth of vegetation was also faster under haying than grazing. Rapid regrowth of uniform vegetation resulted in higher values of EVI within a month of hay harvest. However, regrowth of both hayed and grazed pastures will depend on precipitation after haying/grazing. This difference in effects on vegetation phenology due to grazing and haying will affect carbon and water budgets in tallgrass prairie. Carbon gain increased rapidly due to regrowth of uniform vegetation after haying of a planted pasture because of higher photosynthetic capacity of regrowth (i.e., new) vegetation [11]. Older and more-mature leaves tend to have lower rates of photosynthesis due to decay in cellular mechanisms and function of photosynthetic mechanisms.

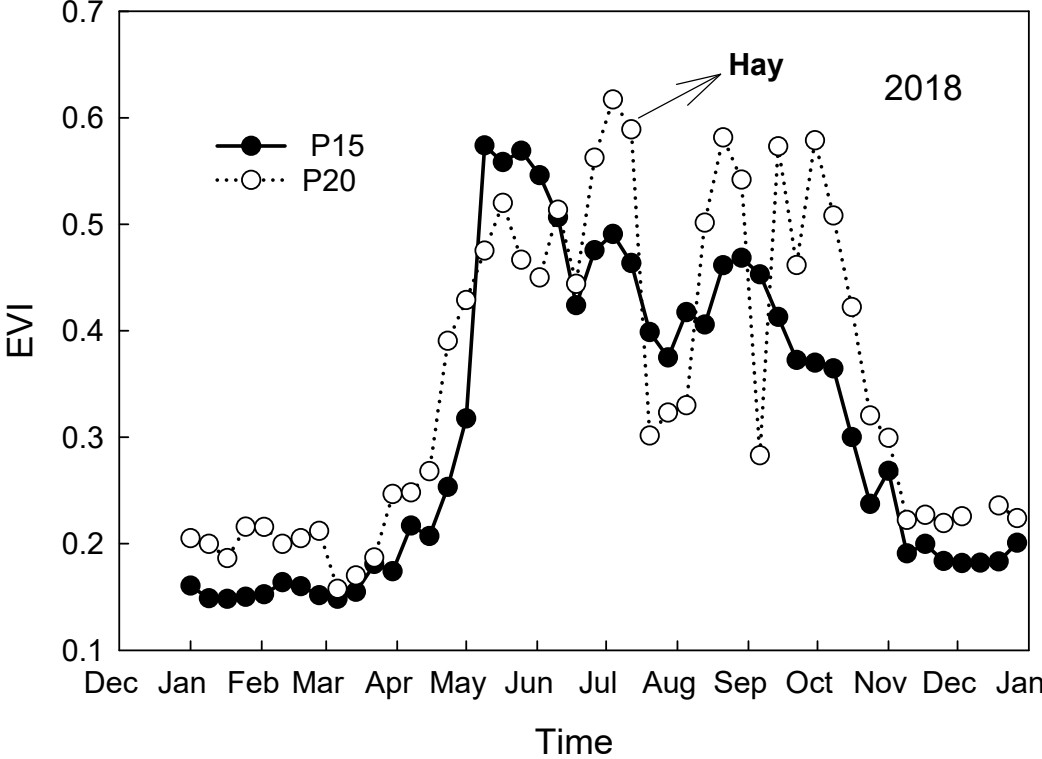

**Figure 7.** Seasonal dynamics of the enhanced vegetation index (EVI) for two pastures with different management (i.e., grazing vs. haying). Pasture 15 was grazed and pasture 20 was hayed in early July.

### 3.6. Impact of Grazing Management on Vegetation Phenology

Figure 8 shows that the EVI continued increasing (as high as 0.6) in P13 though the pasture was grazed during June–July in 2015 with stocking rate of 0.39 head·ha$^{-1}$ (average cattle weight of 480 kg). This increase most likely occurred due to new vegetative growth, facilitated by good amount of precipitation, was higher than the removal of biomass by cattle grazing. However, the EVI plateaued at around 0.4 or declined when the pasture was grazed from May to mid-June 2016 with a stocking rate of ~0.7 head·ha$^{-1}$ (average cattle weight of 567 kg). This response indicated that new vegetation growth, hampered by poor precipitation, did not exceed the amount of biomass removed by grazing cattle when the stocking rate was substantially higher. The results indicate the confounding effects of several factors (i.e., cattle size, stocking rate, precipitation) on vegetation phenology.

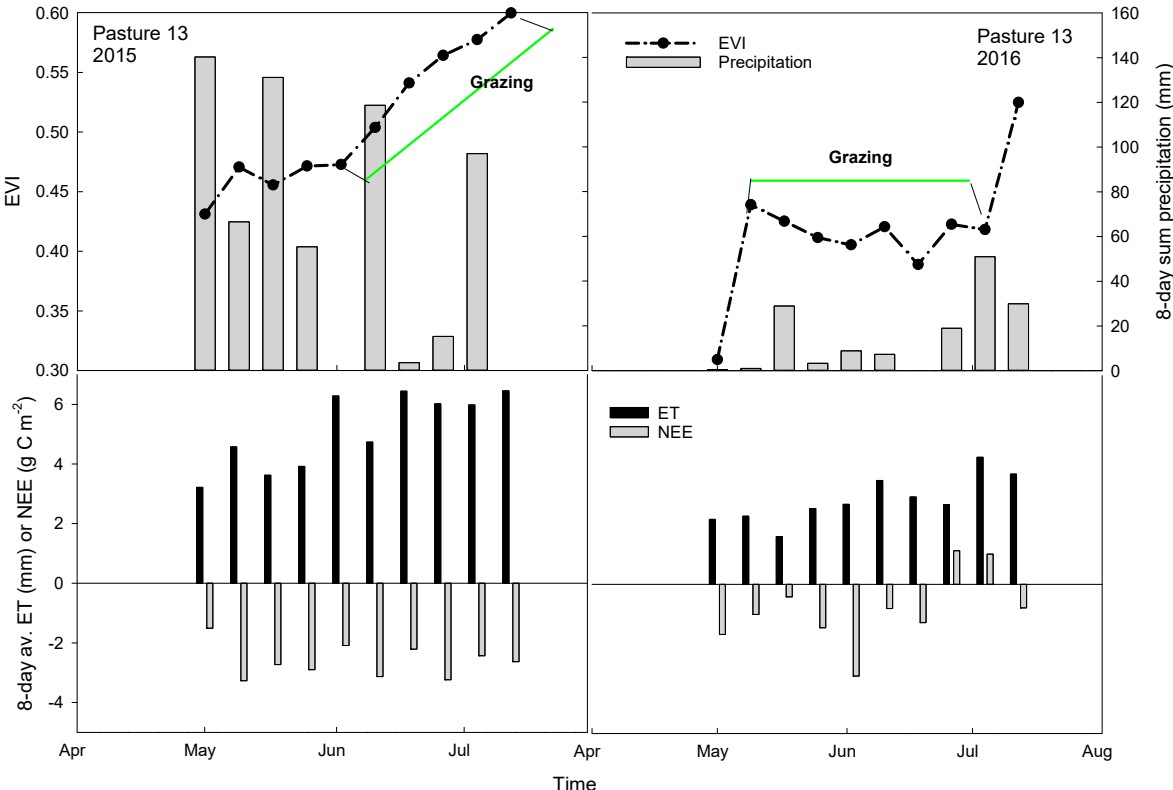

**Figure 8.** Patterns of the enhanced vegetation index (EVI), 8-day sum precipitation, and 8-day average net ecosystem $CO_2$ exchange (NEE) and evapotranspiration (ET) for pasture 13 in 2015 and 2016. Negative NEE indicates sink of carbon and positive NEE indicates source of carbon.

*3.7. Role of Vegetation Phenology on Eddy Fluxes*

The magnitudes of vegetation phenology (i.e., EVI) showed strong correspondence with the magnitudes of eddy fluxes (NEE and ET, Figure 8). Like EVI, the magnitudes of NEE and ET did not decline during grazing in 2015. The maximum 8-day average ET approached 6.4 mm $d^{-1}$ and NEE reached −3.2 g C $m^{-2}$ (sink of carbon) during grazing in 2015. As with EVI, the magnitudes of ET and NEE in 2016 were also smaller during grazing, as 8-day average ET was generally <3–4 mm $d^{-1}$ and the pasture was near carbon neutral. Strong correspondence between eddy fluxes and EVI suggests that changes in vegetation phenology or ecosystem structures with respect to management practices will influence carbon and water budgets in tallgrass prairie. These results were consistent with the findings of many studies that reported strong correspondence between seasonality of satellite-derived vegetation indices (e.g., EVI) and EC-measured/derived fluxes (e.g., ET and GPP) in tallgrass prairie [11,35–37]. The strong correspondence between eddy fluxes and EVI resulted in strong linear relationships of EVI with ET and GPP (Figure 9). The result indicated that the EVI alone explained 75% and 83% of variations in ET in pastures 13 and 18, respectively. Owing to such strong relationships between EVI and ET in tallgrass prairie, Wagle et al. [35] developed a statistical model (ET = 0.11 PAR + 5.49 EVI − 1.43, $R^2$ = 0.86) using data from multiple AmeriFlux tallgrass prairie sites in the central United States to predict daily ET at 8-day intervals.

There was also a strong relationship between EVI and GPP (Figure 9). However, the EVI at finer scales of spatial resolution showed stronger relationships, most likely due to less heterogeneity within the pixels (i.e., reduction in numbers of mixed pixels). Pasture 18 was divided into nine different paddocks that were grazed at different times during May to July, resulting in more heterogeneous conditions within the pasture. Landsat-derived EVI extracted for the paddock which contained the flux tower had the strongest relationship, followed by MODIS-derived EVI at 250 and 500 m spatial resolutions. The results illustrate that mismatch on spatial resolution of remote sensing and

EC footprints, and spatial heterogeneity of the flux measured area, can introduce uncertainties in modeling/upscaling of ground-measured eddy fluxes at large spatial scales using coarser spatial resolution remote sensing data.

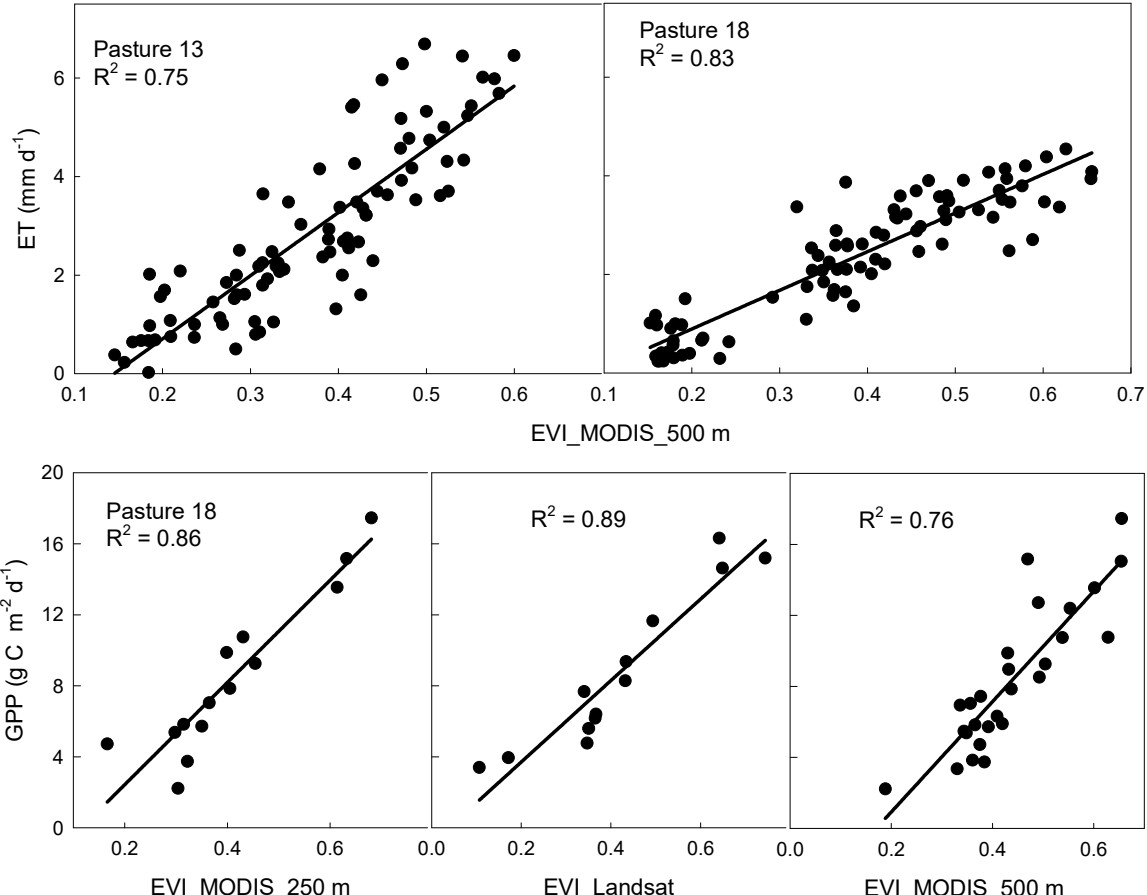

**Figure 9.** Regressions of evapotranspiration (ET) and gross primary production (GPP) with Landsat-derived and Moderate Resolution Imaging Spectroradiometer (MODIS)-derived enhanced vegetation index (EVI) for pastures 13 and 18. The Landsat-derived EVI calculated for the paddock containing flux tower was used in the regression.

## 4. Summary and Future Directions

Early green-up of vegetation and higher forage yields in burned than unburned pastures indicated the importance of burning in tallgrass prairie. However, prairie pastures should not be burned during dry (drought) years as burning further reduces availability of soil water. Distribution of precipitation during growing seasons should be considered more importantly when estimating forage availability rather than annual or seasonal precipitation. Tallgrass prairie pastures in different landscape positions responded differently to management practices, especially in dry years. Impacts of grazing also differed with timing and duration of grazing, cattle size, stocking rate, and precipitation. Thus, several factors related to grazing management (e.g., grazing intensity, duration, and timing) and climatic conditions should be considered when assessing the impact of grazing on forage availability. In addition, tallgrass prairie systems can show inconsistent responses to management practice(s) due to interacting effects of multiple management practices and inter-annual climatic variability. Strong correspondence between eddy fluxes and remotely-sensed EVI suggest that the impacts of management practices on vegetation phenology or ecosystem structures will have implications on carbon and water dynamics in tallgrass prairie systems.

To address the inconsistent responses of tallgrass prairie systems to management practice(s), this long-term experiment will continue to collect EC-measured ET and $CO_2$ fluxes, biometric measurements, vegetation phenology, and climate data from tallgrass prairie systems with different burning, grazing, land use (haying vs. different grazing regimes) treatments, and timing of applied treatments in different landscape positions over a series of variable climatic years. Such data will allow us to undertake a series of comparisons of prairie responses to develop a holistic approach in defining best management practices for this important land use type in the Southern Great Plains. The overall long-term extensive monitoring of prairie systems will also aid collaborative research and modeling efforts across wider disciplines.

In addition to comparing responses across differently managed pastures, understanding variability in forage production, macronutrient availability, soil and landscape features, cattle grazing behavior, and forage utilization within a pasture allows us to deal with them using alternative strategies to further improve the management of pastures. Thus, this long-term experiment will consider the following major areas in the future:

### 4.1. Monitoring Cattle Behavior and Pasture Utilization

Most animal behavior studies have been performed in small pastures. However, cattle in small, familiar, and intensively managed pastures may behave differently than in extensive rangelands [38]. Cattle behavior characteristics and pasture utilization will be assessed more precisely using global positioning system (GPS) and geographic information system (GIS) technologies within the larger production-scale pastures (P13–P16). Lightweight GPS collar receivers will be used to monitor cattle positions at short intervals. This data will allow us to assess the effect of several factors such as how landscape features, forage composition and availability, location of shade/water/supplements, and environmental conditions affect cattle behavior and movement, and use of pasture areas.

### 4.2. Identifying Spatial Variations within a Pasture

Optimizing the production and quality of forage requires consideration of spatial components of pastures [39]. Plant species diversity, functional composition, and forage production vary by landscape positions [39]. Hence, pastures will be divided into multiple zones based on landscape positions, and location and distance from shade/water/supplements. These multiple zones within a pasture will be monitored over time using high-resolution satellite imagery and/or images taken from small unmanned aerial systems (UAS). Plant species composition, biomass yield, LAI, soil types, soil profile moisture, microbial community, water quality, and greenhouse gas emissions (e.g., soil fluxes of nitrous oxide and $CO_2$) will be sampled at monthly intervals during the active growing season from these multiple zones.

Re-distribution of macronutrients (e.g., N, P, K, S, Ca, and Mg) in excreta tends to be non-uniform in grazed pastures, and can result in high concentrations within localized areas near lounging places, supplement feeders, shade, and water troughs, thereby affecting pasture heterogeneity and the environment [40–44]. Thus, macronutrient distributions within a pasture will be monitored comprehensively to maximize the efficiency of pastures.

### 4.3. Assessing Environmental Impact on Animal Behavior

Studies have highlighted the influence of environmental factors on animal behavior. For example, air temperature, wind speed, and temperature-humidity index explained 49% of variations in the amount of time cattle remained in shade [45]. The impact of environment on cattle behavior and movement, and pasture utilization will be further studied using weather data (e.g., air temperature, humidity, wind speed, and wind direction) measured at EC stations.

**Author Contributions:** P.W. wrote the manuscript and P.H.G., B.K.N., P.J.S., and J.P.S.N. revised and contributed for intellectual contents.

**Funding:** This research was partly funded by USDA-NIFA's Agriculture and Food Research Initiative (AFRI), grant number No. 2013-69002.

**Acknowledgments:** The authors would like to thank the tremendous support of several USDA-ARS GRL staffs for managing the pastures, establishing experiments, and assisting in data collection. We also thank two anonymous reviewers for their thoughtful comments.

**Conflicts of Interest:** The authors declare no conflict of interest for this study.

**Disclaimers:** "Mention of trade names or commercial products in this publication is solely for the purpose of providing specific information and does not imply recommendation or endorsement by the U.S. Department of Agriculture.""The U.S. Department of Agriculture (USDA) prohibits discrimination in all its programs and activities on the basis of race, color, national origin, age, disability, and where applicable, sex, marital status, familial status, parental status, religion, sexual orientation, genetic information, political beliefs, reprisal, or because all or part of an individual's income is derived from any public assistance program. USDA is an equal opportunity provider and employer."

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
