# Peer review of "Response of Tallgrass Prairie to Management in the U.S. Southern Great Plains: Site Descriptions, Management Practices, and Eddy Covariance Instrumentation for a Long-Term Experiment"

_remotesensing, doi:10.3390/rs11171988_

Round 1
Reviewer 1 Report
Introduction
L48 suggest “… other treatments and or management practices, their timing…”
L83 “…under a variable …”
L88 “…paper is to…”
L89 “…under a variable …”
L90 “…understanding of the…”
Materials and Methods
L99 “experimental”
L100-101 “local easterly” ? confusing wording
L116 “borders”
L136 “… receiving a prescribed…”
L164 “… for the remainder…”
L185 “late-February”
L189-190 does burning always remove patch grazing effects? Is this only from a visible standpoint? It would seem that the heavier grazed areas would have less fuel , thus not as likely to burn and so not affect directly by the fire…. And one could argue still more affected by previous grazing?
L210 remove spaces
L215 are being collected? Have been collected?
L220 can you provide a little more information on the location and orientation of these transects?
Results and Discussion
L225 “temperate continental”?
L267 “unburned”
L269 “unburned”, “non-burned”?
L275 “…during the drought…”
L276 “of a prescribed…”
L277 “… during the 2011…”, “impacts”
L288 “… removing the litter…”
L292 “…well-distributed…”
L296 suggest “…due to increased mineralization from…”
L299 “…precipitation but not during…”
L300 suggest replacing dash with semi colon after “ways”
L319 “… of a prescribed…”
L320 “unburned”?
Summary
L407 “… over a series…”, “…such data will…”
L409 “Southern Great Plains”?
L423 suggest “….will allow us to assess…”
Overall, good concept for the study, reasonably well written. I realize this is preliminary data from a few years, but you should have more meaningful data in a few more years.
Reviewer 2 Report
The MS titled “Response of Tallgrass Prairie to Management in the U.S. Southern Great Plains: Site Descriptions, Management Practices, and Eddy Covariance Instrumentation for a Long-Term Experiment” provided a simple description about a long-term experiment called intergrated Grassland-Livestock Burning Experiment (iGLOBE) and reported some preliminary findings from this experiment. The experiment intends to help ascertain effects of prescribed burn in tallgrass prairie pastures. Eddy covariance approach was used to collect pre- and post-treatment CO2, H2O, and surface energy fluxes data. Different plots were grazed in rotation and during different time of the year. Overall, the paper is well written and easy to follow. In my opinion, this paper is suitable for publication in the Remote Sensing journal. Below are general comments, mostly for clarification in the introduction, for some minor revisions before acceptance.
L39: Please briefly describe the factors that drove/are driving the decline of the tallgrass prairies, as well as current/projected trend of the tallgrass prairies. This will provide context for audience unfamiliar with the system to better understand the significance of the experiment.
L40: Is there any estimates (e.g., in percentage area) to define “some” and “others”? And are the differences in management practice (i.e., L40-43) related to agencies, land ownership, region, or some other factors? Please clarify.
L43-44: Could you provide a citation that suggest the necessity of such prescribed burning? If not, perhaps connect this and the next sentence using “because” as the conjunction.
L48: What are these other treatments/managements that you are referring to here? Please be more specific.
L53: You used “longer” here, but no context has been provided about the duration of previous experiments and monitoring efforts. Please describe the typical duration of previous experiments mentioned on L50-52, and then explain why having a longer period may provide better information to help tease out these effects.
L63-65: This sentence is vague. I would rephrase it to something like, “If the spatiotemporal variability of CO2 and H2O fluxes and the underlying mechanisms can be better characterized, then such information can be extrapolated and applied over broader area and longer periods…”
L66: Reverse the position. “Comparative Study of…is scarce.”
L88: The purpose of this paper is [to]
L104: What is the approximate size of the herd?
Figure 1: What are the black polygons in the north? Please add to legend or remove from figure.
Figure 7: The “hay” arrow is not properly showing the haying period. Consider using some other ways to show the haying period, e.g., vertical lines, or colored box. And briefly describe it to the caption.
Figure 8: Is here a better way to represent the grazing period? Solid vs. dotted line, or different line color perhaps?
